# The Triterpenoids from *Munronia pinnata* and Their Anti-Proliferative Effects

**DOI:** 10.3390/molecules28196839

**Published:** 2023-09-28

**Authors:** Xuerong Yang, Peiyuan Liu, Yulu Wei, Jingru Song, Xiaojie Yan, Xiaohua Jiang, Jianxing Li, Xiangqin Li, Dianpeng Li, Fenglai Lu

**Affiliations:** 1Guangxi Key Laboratory of Plant Functional Phytochemicals and Sustainable Utilization, Guangxi Institute of Botany, Guangxi Zhuang Autonomous Region and Chinese Academy of Sciences, No. 85 Yanshan Road, Guilin 541006, China; yxrxl@sina.cn (X.Y.); aysrdlc20741@163.com (P.L.); weiyulu0109@126.com (Y.W.); sjr@gxib.cn (J.S.); yxj@gxib.cn (X.Y.); jxh@gxib.cn (X.J.); 2Engineering Research Center of Innovative Traditional Chinese, Zhuang and Yao Materia Medica, Ministry of Education, Guangxi University of Chinese Medicine, No. 13 Wuhe Road, Nanning 530200, China; 3School of Pharmacy, Guilin Medical University, No. 1 Zhiyuan Road, Guilin 541199, China; 4Guangxi Key Laboratory of Plant Conservation and Restoration Ecology in Karst Terrain, Guangxi Institute of Botany, Guangxi Zhuang Autonomous Region and Chinese Academy of Sciences, No. 85 Yanshan Road, Guilin 541006, China; lijx@gxib.cn (J.L.); lxq@gxib.cn (X.L.)

**Keywords:** tirucallane, Meliaceae, *Munronia pinnata*, munropenes A–F, cytotoxic activity

## Abstract

Six new tirucallane-type triterpenoids, named munropenes A–F (**1**–**6**), were extracted from the whole plants of *Munronia pinnata* using a water extraction method. Their chemical structures were determined based on detailed spectroscopic data. The relative configurations of the acyclic structures at C-17 of munropenes A–F (**1**–**6**) were established using carbon–proton spin-coupling constants (^2,3^*J*_C,H_) and inter-proton spin-coupling constants (^3^*J*_H,H_). Furthermore, the absolute configurations of munropenes A–F (**1**–**6**) were determined through high-performance liquid chromatography (HPLC), single-crystal X-ray diffraction, and electronic circular dichroism (ECD) analyses. The antiproliferative effects of munropenes A–F were evaluated in five tumor cell lines: HCT116, A549, HepG2, MCF7, and MDAMB. Munropenes A, B, D, and F (**1**, **2**, **4**, and **6**) inhibited proliferation in the HCT116 cell line with IC_50_ values of 40.90, 19.13, 17.66, and 32.62 µM, respectively.

## 1. Introduction

*Munronia pinnata* (Wall.) W. Theob., also called *M. henryi* Harms, belongs to the family Meliaceae [1,2]. It is a low subshrub that is naturally distributed in several countries, including India, China, the Philippines, and others [3]. In traditional Chinese medicine, *Munronia pinnata* is recognized for its efficacy in treating tuberculosis, cough, stomach pain, and sores [1,2,3,4].

Numerous structurally diverse compounds have been extracted from this plant, including limonoids, triterpenoids, flavonoids, lignans, sterols, sesquiterpenoids, and diterpenoids [3,5,6]. These compounds exhibit a wide range of bioactivities, such as anti-inflammatory, antiproliferative, anti-tobacco mosaic virus, and insect antifeedant activities [3,7,8,9], and have various roles in preserving food, flavoring, and treating various illnesses. In the early stages of our research, phytochemical study on the aerial parts of *Munronia pinnata* was isolated six novel limonoids [10]. In this study, six triterpenoids, named munropenes A–F (**1**–**6**) (Figure 1) were obtained from the whole plants of *Munronia pinnata* using a water extraction method, and their antiproliferative activities against several tumor cell lines, including HCT116, A549, HepG2, MCF7, and MDAMB acquired from the cell bank of Chinese Academy of Sciences (Shanghai, China) were also carried out.

## 2. Results and Discussion

In this investigation, compounds **1** and **2** were obtained as colorless amorphous solids with optical activity: {[α]_D_^20.1^ + 4.98 (*c* 0.10, MeOH) for **1**; [α]_D_^20.1^ − 5.00 (*c* 0.10, MeOH) for **2**}. The results of HRESIMS suggest that compounds 1 and 2 have the same molecular formula—C_32_H_50_O_12_ {*m*/*z* 625.3354 ([M − H]^−^, Δ + 12.4 mmu) for **1** and *m*/*z* 625.3136 ([M − H]^−^, Δ − 9.4 mmu) for **2**}—indicating that both compounds are isomers with eight degrees of unsaturation. The IR spectrum showed the presence of carbonyl functionalities at 1748 cm^−1^ and 1689 cm^−1^ for **1** and **2**. The ^1^H NMR spectrum revealed the existence of one trisubstituted olefin, eight sp^3^ methines, seven sp^3^ methylenes, and six singlet methyls (including one acetyl methyl). The ^13^C NMR spectrum displayed 32 signals, including 3 ester carbonyls, 2 olefinic, 2 oxygenated tertiary, and 3 quaternary carbon signals (Table 1). These data indicate a tetracyclic triterpene structure for compounds **1** and **2**, with the primary difference being the configuration at C-7.

The tetracyclic ring moiety includes an α,β-unsaturated-ε-caprolactone ring (C-1–C-5, C-10) with a formyl group and a methoxy group at C-4, as well as three methyl groups at C-8, C-10, and C-13. The ^1^H-^1^H COSY cross-peaks were observed among H-1/H_2_-2, H-5/H2-6/H-7, H-9-H_2_-11/H_2_-12, and H-15/H_2_-16/H-17. Additionally, HMBC correlations were found between H-1 and C-3; H_2_-28 and C-4, C-5, and C-29; H_3_-30 and C-7, C-8, C-9, and C-14; H_3_-19 and C-1, C-5, C-9, and C-10; H_3_-18 and C-12, C-13, C-14, and C-17; and H-15 and C-14 (Figure 2). From the IR spectrum and the degree of unsaturation of compound **1**, it was determined that an α,β-unsaturated-ε-caprolactone ring is present. Furthermore, the presence of an acetoxy group at C-11 was elucidated by the heteronuclear multiple bond correlation (HMBC) between H-1 and the acetoxy carbonyl carbon. The presence of a 1,4,5,6-tetrahydroxy-6-methyl-heptanol moiety at C-17 was suggested based on the ^1^H-^1^H COSY cross-peaks of H-17/H-20 and H_2_-21/H-20/H_2_-22/H-23/H-24, as well as the HMBC correlations between H_3_-26 and C-24, C-25, and C-26, and H-21 with C-20 and C-22. In short, the planar structure of **1** was established as described.

The relative configuration of the tetracyclic ring moiety in compound **1** was primarily confirmed using the ROESY method (Figure 3). The rotating-frame nuclear Overhauser effect spectroscopy (ROESY) correlations of H_3_-19/H-1, H_3_-19/H-6b, H_3_-19/H_3_-30, H_3_-30/H-7, H_3_-30/H-17, H-16b/H-17, and H-7/H-6β suggested that these protons were cofacial and they were arbitrarily assigned as β-oriented. Consequently, the orientations of H-5, H-16a, H-9, H-18, and H_2_-29 were assigned as α-oriented based on the ROESY correlations between H_2_-28/H-5, H-5/H-6a, H-5/H-9, H-9/H_3_-18, and H_3_-18/H-16*α*. The relative configurations of C-20, C-23, and C-24 were determined based on *J*-based configuration analysis [11]. In addition to the ^3^*J*_H,H_ values, ^2,3^*J*_C,H_ values were detected using hetero half-filtered TOCSY (HETLOC) [12,13], phase-sensitive COSY (PS-COSY) [14], and phase-sensitive HMBC (PS-HMBC) [15,16] spectra of **1** in CD_3_OD. The relative magnitudes of coupling constants assigned from the ^3^*J*_H,H_ and ^2,3^*J*_C,H_ values indicated that each of C-17–C-20, C-20–C-22, C-22–C-23, and C-23–C-24 bonds adopted a single dominant conformer (Figure 4), which was further supported by the ROESY correlations (Figure 3). Thus, the relative configuration at C-20, C-23, and C-24 in compound **1** was assigned as *R**, *S**, and *S**, respectively.

The confirmation was obtained through the utilization of single-crystal X-ray diffraction (Figure 5). Accordingly, the relative configuration of compound **2** was assigned as 1*S**, 4*S**, 5*R**, 7*R**, 8*R**, 9*R**, 10*R**,13*S**, 17*S**, 20*S**, 23*R**, and 24*R** based on the comparison of 1D NMR data (Table 1) and the ^3^*J*_H,H_ and ^2,3^*J*_C,H_ values of compound **2** with those of compound **1**. This determination was further supported by ROESY correlations, such as H-9/H_3_-18 and H_3_-18/H-17, in compound **2** (Figure 3). The absolute configurations of compound **2** were assigned using the electronic circular dichroism (ECD) spectrum since obtaining a crystal for single-crystal X-ray diffraction data was not possible. The TDDFT {CAM-B3LYP/6-31G + (d)} calculation of a possible enantiomer (1*S**, 4*S**, 5*R**, 7*R**, 8*R**, 9*R**, 10*R**,13*S**, 17*S**, 20*S**, 23*R**, 24*R**) of compound **2** yielded a calculated ECD spectrum that matched the experimental spectrum of compound **2** (Figure 6), confirming the 1*S**, 4*S**, 5*R**, 7*R**, 8*R**, 9*R**, 10*R**,13*S**, 17*S**, 20*S**, 23*R**, and 24*R** configurations of compound **2**. Thus, the chemical structures of compounds **1** and **2** were established as shown in Figure 1.

Munropene C (compound **3**) was obtained as an optically active, colorless amorphous solid. The specific rotation [α]_D_^20.0^ = −50.51 (c 0.10, MeOH) indicated its optical activity. From the HRESIMS, a sodiated molecular ion at *m*/*z* 643.3221 ([M − H]^−^, Δ−11.4 mmu) was observed, revealing the molecular formula of compound **3** to be C_32_H_52_O_13_, suggesting the presence of seven degrees of unsaturation. The 1D NMR spectra of compound **3** (Table 1) were similar to those of compound **1**, except for signals related to ring A. By comparing the degrees of unsaturation and molecular formula of **3** with those of compound **1**, it was concluded that compound **3** was a ring A-*seco* munropene A (**1**). This conclusion was further supported by the 1H NMR chemical shifts of H-1 and H-2 in compound **3** (Figure 2). Therefore, a possible biosynthetic pathway for munropene C (compound **3**) was proposed, suggesting that it might be generated through the hydrolysis of munropene A (compound **1**) in ring A (Appendix A).

The β orientation of H-5, H-7, H-17, H_3_-19, and H_3_-30 was assigned based on the ROESY correlations of H_3_-19/H-5, H_3_-19/H_3_-30, H_3_-30/H-7, H_3_-30/H-17, and H-7/H-5. The ROESY cross-peaks of H-9/H_3_-18 suggested the α orientations of H-9 and H_3_-18 (Appendix A). Thus, considering the similar biosynthetic pathway of compounds **1** and **3**, the S* configuration was assigned to C-1 and C-4 of compound **3**. The ROESY cross-peaks of H-1/H-5, H-1/H_3_-19, H_2_-29/H-5, and H_2_-29/H_3_-19 further supported this assignment (Appendix A). Consequently, the structure and relative configuration of compound **3** were established as shown.

The molecular formula of munropene D (compound **4**) was determined to be C_38_H_62_O_14_ through HRESIMS analysis, showing a peak at *m*/*z* 741.4010 ([M − H]^−^, Δ−5.7 mmu). The 1D NMR spectra of compound **4** (Table 1) exhibited signals originating from a glucose group, indicating its structural similarity to that of compound **3**, except for the modifications at the C-5 and C-7 positions. The attachment of a methylethylene moiety at C-5 was confirmed by the HMBC correlation of H_3_-29 with C-4, C-5, and C-28.

The sugar moiety was obtained through acid hydrolysis, followed by treatment with L-cysteine methyl ester and *o*-tolylisothiocyanate, resulting in a reaction mixture that produced a peak during HPLC analysis identical to that of the derivative of authentic D-glucose prepared using the same procedure [17]. Hence, the glucose moiety of compound **4** was determined to be D-glucose. The β-glycosidic linkage of the D-glucosyl moiety at C-7 was concluded based on the coupling constant value of the anomeric proton (H-1′, J = 7.8 Hz), as well as the HMBC correlation of H-1′ with C-7 (Figure 2).

The relative configurations of compound **4** were assigned as 1*S**, 4*S**, 5*R**, 7*R**, 8*R**, 9*R**, 10*R**, 13*S**, 17*R**, 21*R**, 23*S**, and 24*S** through a comparison of the NMR data of compound **4** with that of compound **1**. ROESY cross-peaks observed in compound **4** between H_3_-19/H_3_-30, H_3_-30/H-7, H_3_-30/H-17, H-5/H-9, H_3_-18/H-9, H_3_-18/H-16a, and H-16b/H-17, which were also present in compound **1**, supported this assignment.

Munropenes E (compound **5**) and F (compound **6**) were isolated as optically active colorless amorphous solids. Their optical rotations were determined as [α]_D_^20.0^ = −24.04 (c 0.10, MeOH) for compound **5** and [α]_D_^20.0^ = −6.80 (c 0.10, MeOH) for compound **6**. HRESIMS analysis revealed the molecular formulas as C_36_H_58_O_12_ (*m*/*z* 727.3846 [M + HCOO]^−^, Δ−6.4 mmu for compound **5**; *m*/*z* 727.3845 [M + HCOO]^−^, Δ−6.5 mmu for compound **6**), suggesting the presence of eight degrees of unsaturation. The 1H NMR data (Table 2) displayed resonances corresponding to a trisubstituted olefin, a 1,2-disubstituted olefin, seven sp^3^ methines, seven sp^3^ methylenes, six singlet methyls, and a glucosyl moiety. The ^13^C NMR spectrum exhibited 36 signals, including 1 ketone carbonyl, 4 olefinic, 1 oxygenated tertiary, and 4 quaternary carbon signals (Table 2). These data indicate that compounds **5** and **6** are isomers of each other and closely related to compounds **1** and **2**, except for changes occurring in the A ring and at C-7. The glucose moiety of compounds **5** and **6** was determined to be d-glucose through similar HPLC analyses as performed for compound **4**. The comparison of the 1D NMR spectroscopic data of compounds **5** and **6** with those of compound **4** indicated that the β-glycosidic linkage of the d-glucosyl moiety was attached at C-7, which was confirmed by the HMBC correlation of H-1′ with C-7, as well as similar HPLC analyses as conducted for compound **4**. The A rings of these two compounds were assigned as α, β-unsaturated hexane ketones with one methyl and one methanol group at C-4, elucidated by ^1^H-^1^H COSY cross-peaks of H-1/H-2 and the HMBC correlations of H3-29 with C-3, C-4, C-5, and C-28, as well as H-1 with C-3, C-5, and C-10. Additionally, the HMBC correlations of H_3_-19 with C-1, C-10, C-5, and C-9 supported this assignment and allowed for the connectivity between ring A and ring B.

The relative configurations of compounds **5** and **6** in the aglycone moieties were deduced to be similar to those of compounds **1** and **2**, respectively, based on the resemblance of their 1D NMR data (Table 1 and Table 2) and ROESY correlations (Appendix A). The ECD spectra of compounds **5** and **6** indicated a similar Cotton effect at 237 nm and 203 nm. According to the octant rule [18], the positive Cotton effect observed in compounds **5** and **6**, attributed to the exciton coupling of α, β-unsaturated hexane ketone, suggested the absolute configuration of 4*S**, 5*R**, and 10*R** in compounds **5** and **6**. The absolute configuration of compound **6** was confirmed by comparing the experimental ECD spectrum with the TDDFT calculated spectrum. The experimental ECD spectrum of compound **5** correlated well with the calculated spectrum of a possible enantiomer with the 4*S**, 5*R**, 7*R**, 8*R**, 9*R**, 10*R**, 13*S**, 17*R**, 20*R**, 23*S**, and 24*S** configurations (Figure 7), confirming the assignment of the absolute configuration of compound **5** as mentioned above. Thus, the structures of compounds **5** and **6** were elucidated as shown in Figure 1.

As part of our ongoing search for potential natural product leads for therapeutic agents from *M. pinnata*, we evaluated the antiproliferative activity of munropenes A–F (compounds **1**–**6**) against various human cancer cell lines including HCT116, A549, HepG2, MCF7, and MDAMB. Munropenes A, B, D, and F (compounds **1**, **2**, **4**, and **6**) were not cytotoxic (IC_50_ > 50 μM) to A549, HepG2, MCF7, and MDAMB cells. However, they exhibited moderate cytotoxicity against HCT116 cells, with IC_50_ values of 40.90, 19.13, 17.66, and 32.62 μM, respectively (Table 3). In contrast, munropenes C and E (compounds **3** and **5**) did not exert any cytotoxicity against the tested cell lines (Table 3).

## 3. Materials and Methods

### 3.1. General Experimental Protocols

The Jasco P-1020 polarimeter was used to measure optical rotation. Infrared (IR) spectra were obtained using a Tensor 27 spectrometer and a Nicolet Fourier transform infrared spectrometer (Thermo Fisher, Waltham, MA, USA) with KBr pellets. Circular dichroism (CD) spectra were recorded using a J-810 CD spectrometer. MS spectra were measured using an LC/MS-IT-TOF mass spectrometer. Nuclear magnetic resonance (NMR) spectra were recorded on a Bruker AVANCE III-HD 500 spectrometer with MeOH (δ_H_ 3.30 and δ_C_ 49.0) serving as the internal standard. The countercurrent chromatography (CCC) experiment was conducted using a TBE-300C machine (manufactured by Tauto Biotechnique, located in Shanghai, China). HPLC analysis was performed using an Agilent 1260 InfinityIILC system (Agilent Technologies, Santa Clara, CA, USA). The columns utilized were Agilent Poroshell 120 SB-C18 (4 mm, 4.6 mm × 150 mm, Agilent, Santa Clara, CA, USA), ChromCore 120-C18 (5 mm, 10 mm × 250 mm, NanoChrom, Suzhou, China), and Agilent ZORBAX SB-C18 (5 mm, 9.4 mm × 250 mm, Agilent, Santa Clara, CA, USA). Silica gel (200–300 mesh, Qingdao Marine Chemical Factory, Qingdao, China) and MCI gel (Mitsubishi Chemical Corporation, Tokyo, Japan) were used for column chromatography. Thin-layer chromatography (TLC) analyses were performed using preloaded silica gel 60 F254 plates from Merck Millipore in Germany. The spots were visualized by heating the silica gel plate, which was sprayed with a mixture of 10% H_2_SO_4_ and ethanol.

### 3.2. Plant Material

Botanical samples of *Munronia pinnata* (Wall.) W. Theob. were collected in July 2021 from Jingxi City, located in the Guangxi Zhuang Autonomous Region. The plant material was identified by one of the authors, X.-Q. Li. Voucher specimens have been preserved at the herbarium of the Center for Natural Products Chemistry Studies, Guangxi Institute of Botany, Guangxi Zhuang Autonomous Region, and Chinese Academy of Sciences (21-GX-001).

### 3.3. Extraction and Isolation

*Munronia pinnata* (25 kg), which had been air-dried and powdered, was extracted three times with 95% ethanol (250 L) under reflux conditions. The resulting mixture was then filtered to remove any insoluble components. The filtrate was concentrated under reduced pressure to obtain the extract. The extract was further extracted using petroleum ether and EtOAc, yielding a remaining water layer. To obtain fraction Fr 3 (34 g), the water layer was subjected to macroporous resin column chromatography with elution using 20%, 40%, and 80% ethanol. Additionally, the 80% fraction was subjected to gel column chromatography with methanol elution, yielding fraction Fr2 (15 g). Fr2 was separated by C_18_ column chromatography with a methanol gradient elution (MeOH-H_2_O, 30:70–50:50 gradient system) to obtain seven fractions (Fr2.1–Fr2.7). Fr2.1 (1.46 g) was further separated by HSCCC [CH_2_Cl_2_-MeOH-H_2_O (2:2:1, v/v/v)] followed by silica gel (CH_2_Cl_2_-MeOH, 4:1) to yield **2** (68 mg). Fr2.3 (1.27 g) was obtained by HSCCC [CH_2_Cl_2_-MeOH-H_2_O (2:2:1, v/v/v)]. Fr2.3.1–Fr2.3.3. Fr2.3.3 was subjected to prep-HPLC [MeCN-H_2_O-HCOOH (23:77:0.1, v/v/v)] to obtain compound **3** (26 mg, t_R_ = 10.1 min). Fr2.4 (1.39 g) was isolated by HSCCC [CH_2_Cl_2_-MeOH-H_2_O (2:2:1, v/v/v)], and then compound **5** (10 mg, t_R_ = 12.2 min) and compound **6** (15 mg, t_R_ = 14.7 min) were obtained by prep-HPLC [MeCN-H_2_O-HCOOH (23:77:0.1, v/v/v)]. Fr2.7 (0.9 g) yielded compound **4** (184 mg) through HSCCC [CH_2_Cl_2_-MeOH-H_2_O (2:2:1, v/v/v)]. Fr.3 (30 g) was subjected to silica gel column chromatography (CH_2_Cl_2_-MeOH, 80:20–100:0 gradient system) resulting in twenty fractions (Fr3.1–Fr3.20). Fr3.3 (1.49 g) was purified by prep-HPLC [MeCN-H_2_O-HCOOH (18:82:0.1, v/v/v)] to obtain compound **1** (321 mg, t_R_ = 21.3 min).

#### 3.3.1. Munropene A (Compound **1**)

Colorless amorphous solid; [a]_D_^20.1^ + 4.98 (*c* 0.10, 90% MeOH aq.); IR (KBr) n_max_ 3368 (-OH), 2968 (-CH), 1724 (-C=O), and 1636 (-C=C-) cm^−1^; UV (MeOH) λ_max_ 209 (e = A/CL, 24,281) nm; ^1^H and ^13^C NMR (CD_3_OD/D_2_O, Table 1); HRESIMS *m*/*z* 625.3354 ([M − H]^−^, calcd for C_32_H_49_O_12_, 625.3230).

#### 3.3.2. Munropene B (Compound **2**)

Colorless amorphous solid; [a]_D_^20.1^ − 5.00 (*c* 0.10, 90% MeOH aq.); IR (KBr) n_max_ 3422 (-OH), 2930 (-CH), 1720 (-C=O), and 1620 (-C=C-) cm^−1^; UV (MeOH) λ_max_ 197 (e = A/CL, 10,532) nm; ECD (MeOH) De (nm) + 6.0 (223); ^1^H and ^13^C NMR (CD_3_OD/D_2_O, Table 1); HRESIMS *m*/*z* 625.3136 ([M − H]^−^, calcd for C_32_H_49_O_12_, 625.3230).

#### 3.3.3. Munropene C (Compound **3**)

Colorless amorphous solid; [a]_D_^20.0^ − 50.51 (*c* 0.10, 90% MeOH aq.); IR (KBr) n_max_ 3422 (-OH), 2930 (-CH), 1731 (-C=O), and 1630 (-C=C-) cm^−1^; UV (MeOH) λ_max_ 195 (e = A/CL, 13,859) nm; ECD (MeOH) De (nm) + 9.1 (227), +0.5 (197); ^1^H and ^13^C NMR (CD_3_OD/D_2_O, Table 1); HRESIMS *m*/*z* 643.3221 ([M − H]^−^, calcd for C_32_H_51_O_13_, 643.3335).

#### 3.3.4. Munropene D (Compound **4**)

Colorless amorphous solid; [a]_D_^20.0^ − 76.15 (*c* 0.10, 90% MeOH aq.); IR (KBr) n_max_ 3419 (-OH), 2927 (-CH), 1722 (-C=O), and 1636 (-C=C-) cm^−1^; UV (MeOH) λ_max_ 195 (e = A/CL, 26,334) nm; ECD (MeOH) De (nm) + 1.2 (198), −6.3 (211); ^1^H and ^13^C NMR (CD_3_OD/D_2_O, Table 1); HRESIMS *m*/*z* 741.4010 ([M − H]^−^, calcd for C_38_H_61_O_14_, 741.4067).

#### 3.3.5. Munropene E (Compound **5**)

Colorless amorphous solid; [a]_D_^20.0^ − 24.04 (*c* 0.10, 90% MeOH aq.); IR (KBr) n_max_ 3421 (-OH), 2926 (-CH), 1722 (-C=O), and 1650 (-C=C-) cm^−1^; UV (MeOH) λ_max_ 196 (e = A/CL, 12,066) nm; ECD (MeOH) De (nm) + 22.9 (235), −16.8 (204); ^1^H and ^13^C NMR (CD_3_OD/D_2_O, Table 1); HRESIMS *m*/*z* 727.3846 ([M + HCOO]^−^, calcd for C_37_H_59_O_14_, 727.3910).

#### 3.3.6. Munropene F (Compound **6**)

Colorless amorphous solid; [a]_D_^20.0^ − 6.80 (*c* 0.10, 90% MeOH aq.); IR (KBr) n_max_ 3412 (-OH), 2926 (-CH), 1722 (-C=O), and 1656 (-C=C-) cm^−1^; UV (MeOH) λ_max_ 197 (e = A/CL, 13,228) nm; ECD (MeOH) De (nm) + 23.8 (235), −18.1 (203); ^1^H and ^13^C NMR (CD_3_OD/D_2_O, Table 1); HRESIMS *m/z* 727.3845 ([M + HCOO]^−^, calcd for C_37_H_59_O_14_, 727.3910).

### 3.4. Acid Hydrolysis and Sugar Analysis of Munropenes D–F (Compounds ***4***–***6***)

Compounds **4**–**6** (1.5 mg each) were subjected to hydrolysis using 2.0 M HCl (2.0 mL) for a duration of 2 h at a temperature of 90 °C. To establish neutral conditions, anion exchange resin (IRA 400) was added and subsequently removed through filtration. The resulting filtrate was then subjected to vacuum concentration and dried under vacuum conditions. The resultant residue was dissolved in pyridine (1.0 mL) supplemented with L-cysteine methyl ester hydrochloride (1.0 mg) and heated at 60 °C for 1 h. Subsequently, *o*-torylisothiocyanate (1.0 mg) was added to the mixture, which was then stirred at 60 °C for an additional hour. Reversed-phase HPLC was used to directly analyze the reaction mixture, and the retention times of reference compounds and carbohydrate derivatives were compared, which was performed under the following conditions: detection wavelength of 250 nm, mobile phase consisting of 25% acetonitrile–water with 0.1% formic acid, and utilizing an Agilent Poroshell 120 SB-C18 column (4 mm, 4.6 mm × 150 mm, Agilent, Santa Clara, CA, USA). The absolute conformation of the sugar moiety was ascertained through comparison with D-glucose (t_R_ = 9.55 min).

### 3.5. Cytotoxicity Assay

The cytotoxicity of munropenes A–F (**1**–**6**) in A549, HepG2, HCT116, MCF7, and MDAMB was tested using the Cell Counting Kit-8 (CCK-8). A 100 μL cell suspension (2 × 10^5^ cells/mL) was seeded into 96-well plates. Following incubation for 24 h, the cells were treated with various concentrations (5, 10, 20, 40, 80, or 160 µM) of each specific compound, while the control cells received an equal volume of DMSO. Subsequently, after an additional 24 h of culture, 10 μL CCK-8 was added and incubated for an additional 2 h. The absorbance value at 450 nm was detected using a microplate reader, enabling the calculation of the cell survival rate.

## 4. Conclusions

The phytochemical study on the whole plants of a Chinese traditional medicine plant *Munronia pinnata* (Meliaceae) led to the isolation of six new tirucallane-type triterpenoids, munropenes A–F (compounds **1**–**6**). Tirucallane-type triterpenoids are known as major components of plants belonging to Meliaceae, but they had not been systematically studied in *M. pinnata*. In the present paper, munropenes A and B (**1** and **2**) were identified as tirucallane-type triterpenoids with an α,β-unsaturated-ε-caprolactone moiety in ring A, while munropenes C and D (compounds **3** and **4**) were categorized as ring A *seco*-tirucallane-type triterpenoids. Additionally, munropenes D, E, and F (compounds **4**, **5**, and **6**) were determined to be glycosides of tirucallane-type triterpenoids based on 1D and 2D-NMR, HR-ESI-MS, IR, single-crystal X-ray diffraction, ECD, and J-based configuration analyses. Munropenes A, B, D, and F (compounds **1**, **2**, **4**, and **6**) was moderately cytotoxic to the HCT116 cell line, but did not show any cytotoxicity in the A549, HepG2, MCF7, and MDAMB cell lines. Furthermore, munropenes C (compound **3**) and E (compound **5**) exerted no cytotoxicity against all tested cell lines, including HCT116, A549, HepG2, MCF7, and MDAMB cells.

## Figures and Tables

**Figure 1 molecules-28-06839-f001:**
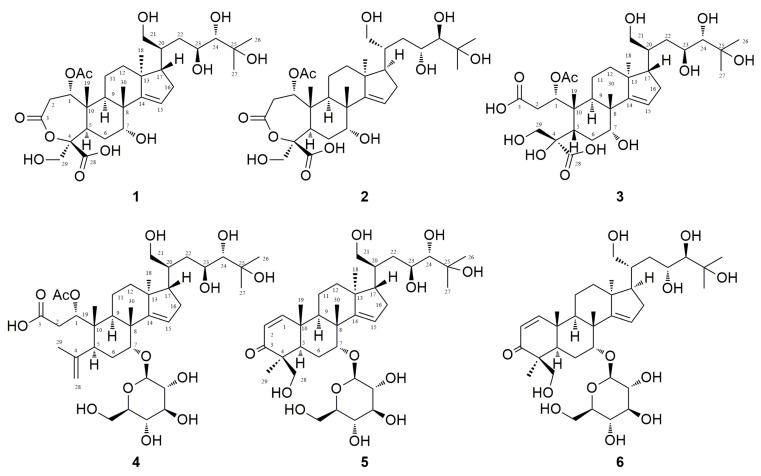
The chemical structures of munropenes A–F (**1**–**6**).

**Figure 2 molecules-28-06839-f002:**
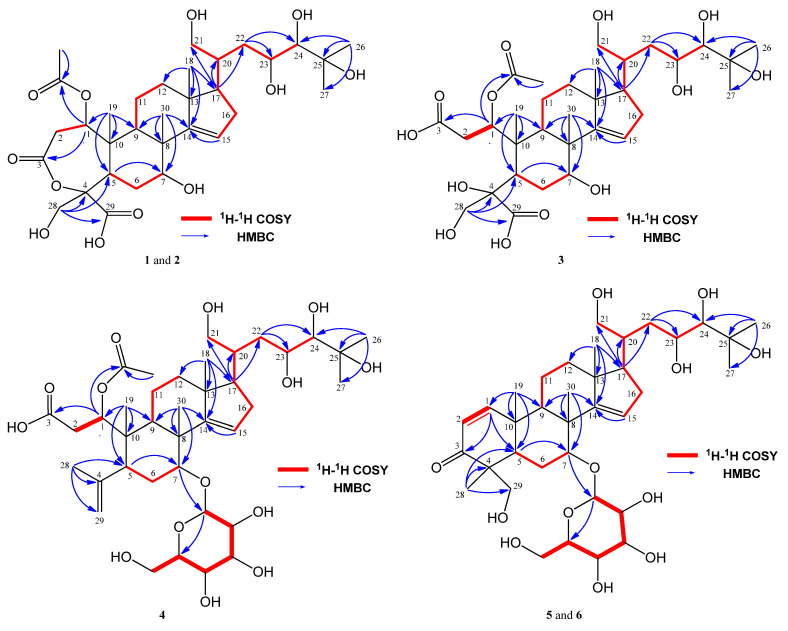
Selected 2D NMR correlations for munropenes A–F (**1**–**6**).

**Figure 3 molecules-28-06839-f003:**
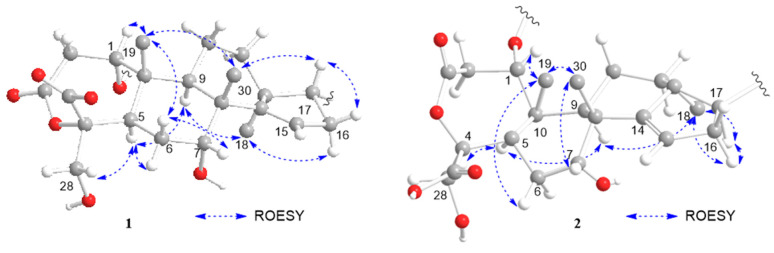
Key ROESY correlations and relative configuration for munropenes A, B (**1**, **2**) (protons of methyl groups are omitted).

**Figure 4 molecules-28-06839-f004:**
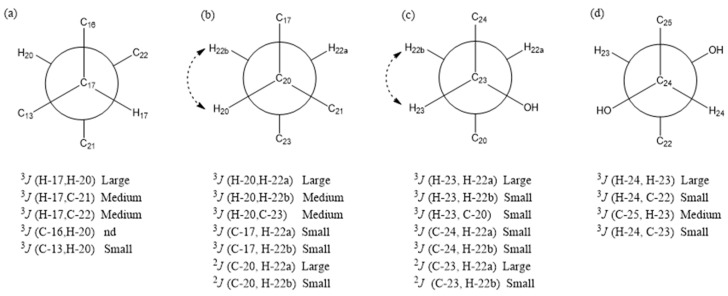
Rotation models for the (**a**) C-17–C-20, (**b**) C-20–C-22, (**c**) C-22–C-23, and (**d**) C-23–C-24 bonds of munropene A (**1**). “nd” means that the magnitude was not determined. Dashed arrows indicate ROESY correlations.

**Figure 5 molecules-28-06839-f005:**
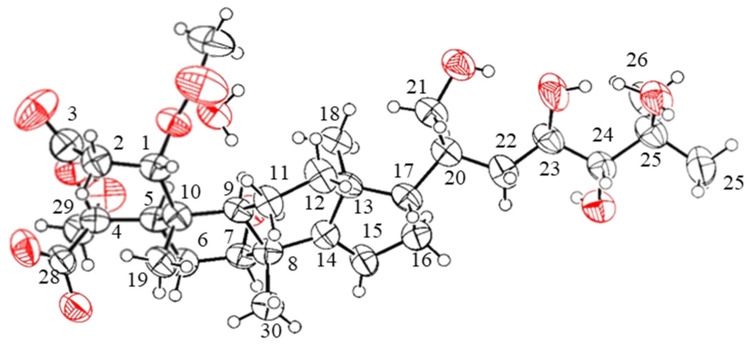
X-ray crystal structure of munropene A (**1**).

**Figure 6 molecules-28-06839-f006:**
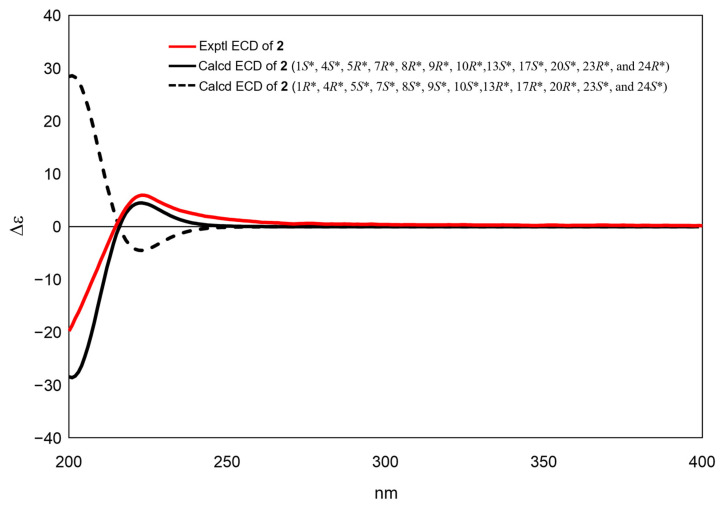
Experimental and calculated ECD spectra of munropene B (**2**).

**Figure 7 molecules-28-06839-f007:**
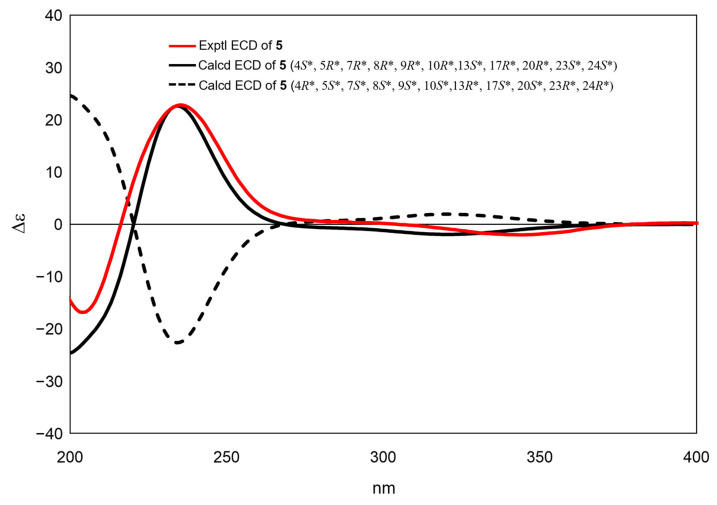
Experimental and calculated ECD spectra of **5**.

**Table 1 molecules-28-06839-t001:** ^1^H and ^13^C NMR data for munropenes A–D (**1**–**4**) in CD_3_OD.

Position	1		2		3		4	
	δ_H_ (*J* in Hz)	δ_C_	δ_H_ (*J* in Hz)	δ_C_	δ_H_ (*J* in Hz)	δ_C_	δ_H_ (*J* in Hz)	δ_C_
1	4.66 (1H, d, 7.2)	71.3	4.78 (1H, d, 7.5)	73.3	6.21 (1H, d, 7.5)	78.8	5.55 (1H, dd, 11.1, 1.6)	79.1
2	3.31 (1H, d,16.0)/ 2.77 (1H, m)	35.5	3.84 (1H, m)/ 2.93 (1H, m)	37.1	2.41 (1H, m)/ 2.99 (1H, m)	36.2	2.41 (1H, dd, 15.1, 11.0)/ 3.08 (1H, m)	36.3
3	-	170.5	-	175.8	-	175.5	-	175.5
4	-	87.4	-	91.4	-	81.9	-	147.7
5	2.77 (1H, m)	39.2	2.83 (1H, dd, 13.0, 2.6)	40.3	2.37 (1H, dd, 13.0, 2.6)	42.7	2.79 (1H, dd, 13.0, 3.4)	43.9
6	2.15 (1H, m)/ 1.87 (1H, m)	27.6	2.49 (1H, m)/ 1.99 (1H, m)	28.8	2.29 (1H, m)/ 1.68 (1H, m)	28.1	2.07 (1H, m)/ 1.76 (1H, dt, 14.5, 3.4)	28.4
7	3.75 (1H, m)	70.7	3.91 (1H, t, 3.1)	73	3.85 (1H, t, 3.1)	73	4.03 (1H, dd, 3.7, 1.5)	78.4
8	-	42.8	-	44.5	-	44.6	-	44.3
9	2.40 (1H, m)	33.3	2.52 (1H, m)	34.8	2.29 (1H, m)	35.3	2.30 (1H, m)	35.5
10	-	43.6	-	45.2	-	47.1	-	45.5
11	1.22 (1H, m)/ 1.40 (1H, m)	16.4	1.47 (1H, m)/ 1.52 (1H, m)	17.8	1.68 (1H, m)/ 2.01 (1H, m)	20.3	1.63 (1H, m)/ 1.94 (1H, m)	20.1
12	1.62 (1H, m)/ 1.40 (1H, m)	34.4	1.84 (1H, m)/ 1.49 (1H, m)	35.6	1.50 (1H, dt, 13.1, 9.3)/ 1.92 (1H, m)	36.1	1.57 (1H, m)/ 1.87 (1H, m)	37.3
13	-	46	-	47.7	-	47.3	-	47.3
14	-	160.2	-	162	-	161.8	-	159.5
15	5.32 (1H, m)	119.1	5.46 (1H, dd, 3.5, 1.5)	121	5.45 (1H, dd, 3.8, 1.5)	120.7	5.45 (1H, dd, 3.7, 1.7)	121.6
16	2.02 (1H, m)/ 2.15 (1H, m)	34.6	2.18 (1H, m)/ 2.34 (1H, m)	35.9	2.18 (1H, m)/ 2.29 (1H, m)	35.9	2.08 (1H, m)/ 2.30 (1H, m)	36.2
17	1.57 (1H, m)	55.9	1.66 (1H, td, 10.4, 7.2)	56.2	1.68 (1H, m)	57.9	1.63 (1H, m)	58.2
18	0.90 (3H, s)	18.4	1.04 (3H, s)	19	1.10 (3H, s)	19.4	1.12 (3H, s)	21.1
19	0.94 (3H, s)	13.1	1.14 (3H, s)	14.1	1.04 (3H, s)	14.8	0.97 (3H, s)	15.7
20	1.68 (1H, m)	39.5	1.84 (1H, m)	41.7	1.19 (1H, m)	41.8	1.87 (1H, m)	42.1
21	3.23 (1H, m)/ 3.59 (1H, m)	63.9	3.34 (1H, m)/3.84 (1H, m)	66	3.36 (1H, m)/ 3.89 (1H, dd, 10.6, 3.6)	66.1	3.34 (1H, m)/ 3.89 (1H, dd, 10.5, 3.6)	66.3
22	1.94 (1H, m)/ 1.22 (1H, m)	36.9	1.36 (1H, m)/ 2.18 (1H, m)	38.5	1.37 (1H, m)/ 2.18 (1H, m)	38.7	1.35 (1H, ddd, 14.7, 9.3, 7.3)/ 2.22 (1H, dt, 14.8, 2.9)	38.8
23	3.49 (1H, m)	72.9	3.66 (1H, td, 8.9,2.1)	74.7	3.67 (1H, td, 8.9, 2.1)	74.8	3.67 (1H, td, 9.0, 2.1)	74.9
24	2.89 (1H, d, 8.1)	78.4	3.09 (1H, d, 8.2)	79.9	3.08 (1H, d, 8.2)	79.9	3.08 (1H, m)	79.9
25	-	73.2	-	74.9	-	75	-	75
26	1.08 (3H, s)	28	1.22 (3H, s)	27.9	1.22 (3H, s)	28	1.24 (3H, s)	23.9
27	1.09 (3H, s)	24	1.24 (3H, s)	24.1	1.24 (3H, s)	23.9	1.22 (3H, s)	28
28	3.62 (1H, m)/ 3.81 (1H, d, 10.9)	172.2	3.78 (1H, d, 14.7)/ 4.01 (1H, d, 11.1)	177.6	3.78 (1H, d, 10.4)/ 4.01 (1H, d, 10.4)	178.7	4.96 (1H, d, 2.4)/ 4.88 (1H, d, 2.4)	116.1
29	-	70.7	-	70.9	-	69.4	1.83 (3H, s)	23.7
30	1.03 (3H, s)	27.4	1.16 (3H, s)	28.3	1.08 (3H, s)	27.6	1.17 (3H, s)	27.5
1-OAc	-	169.5	-	171.6	-	172.5	-	172.4
	1.99 (3H, s)	20.7	2.06 (3H, s)	20.9	2.00 (3H, s)	21.2	2.00 (3H, s)	21.1
1′	-	-	-	-	-	-	4.30 d (7.8)	100.3
2′	-	-	-	-	-	-	3.12 (1H, m)	75.4
3′	-	-	-	-	-	-	3.33 (1H, m)	78.8
4′	-	-	-	-	-	-	3.17 (1H, m)	72.3
5′	-	-	-	-	-	-	3.18 (1H, dd, 9.7, 2.4)	77.6
6′	-	-	-	-	-	-	3.59 (1H, dd, 11.5, 5.9)/ 3.88 (1H, dd, 11.4, 2.3)	63.5

**Table 2 molecules-28-06839-t002:** ^1^H and ^13^C NMR data for munropenes E–F (**5**–**6**) in CD_3_OD.

Position	5		6	
	δ_H_ (*J* in Hz)	δ_C_	δ_H_ (*J* in Hz)	δ_C_
1	7.28 (1H, d, 10.2)	161.3	7.28(1H, d, 10.2)	161.3
2	5.77 (1H, d, 10.2)	125.6	5.77(1H, d, 10.2)	125.6
3		206.3		206.3
4		51.7		51.7
5	2.63 (1H, dd, 10.2, 5.2)	46	2.63(1H, dd, 10.2, 5.4)	46
6	2.03 (1H, m)	23.3	2.03 (1H, m)	23.3
7	4.14 (1H, t, 2.8, 2.8)	78.3	4.14 (1H, t, 2.8, 2.8)	78.4
8		44.9		44.9
9	2.25 (1H, m)	40.1	2.25 (1H, m)	40.1
10		41.1		41
11	1.68 (1H, m)/1.87 (1H, m)	18.7	1.66 (1H, m)/1.87 (1H, m)	18.7
12	1.68 (1H, m)/1.87 (1H, m)	37.3	1.66 (1H, m)/1.87 (1H, m)	37.5
13		47.6		47.6
14		158.7		159.1
15	5.49 (1H, dd, 3.7, 1.7)	122.2	5.49 (1H, dd, 3.7, 1.7)	122.2
16	2.03(1H, m)/2.25(1H, m)	36.1	2.07 (1H, m)/2.30 (1H, ddd, 15.1, 7.2, 3.5)	36.3
17	1.72(1H, m)	57.5	1.66 (1H, m)	58.1
18	1.06 (3H, s)	21.2	1.06 (3H, s)	21.1
19	1.24 (3H, s)	19.7	1.23 (3H, s)	19.7
20	1.80 (1H, m)	40.9	1.86 (1H, m)	42
21	3.44 (1H, dd, 10.6, 6.8)/3.79 (1H, dd, 10.6, 6.8)	65.3	3.35 (1H, m)/3.87 (1H, m)	66.2
22	1.65 (1H, m)/1.77 (1H, m)	38.2	1.34 (1H, m)/2.20 (1H, m)	38.8
23	4.09 (1H, m)	71.2	3.66 (1H, m)	74.9
24	3.17 (1H, m)	78.7	3.08(1H, m)	79.9
25		74.7		75
26	1.27 (3H, s)	27.2	1.22 (3H, s)	28
27	1.24 (3H, s)	26.5	1.24 (3H, s)	23.9
28	3.62 (1H, d, 11.3)/3.74 (1H, d, 11.3)	66.2	3.62 (1H, m)/3.74 (1H, d, 11.4)	66.2
29	1.25 (3H, s)	21.6	1.25 (3H, s)	21.6
30	1.21 (3H, s)	28.3	1.20 (3H, s)	28.2
1′	4.32 (1H, d, 7.6)	100.7	4.32 (1H, d, 7.6)	100.7
2′	3.08 (1H, m)	75.3	3.09 (1H, m)	75.3
3′	3.34 (1H, m)	78.4	3.32 (1H, m)	74.9
4′	3.15 (1H, m)	72.3	3.15 (1H, m)	72.3
5′	3.23 (1H, ddd, 9.2, 6.4, 2.5)	77.6	3.22 (1H, ddd, 9.2, 6.6, 2.4)	77.6
6′	3.35 (1H, m)/3.88 (1H, m)	63.6	3.58 (1H, m)/3.87 (1H, m)	63.6

**Table 3 molecules-28-06839-t003:** IC_50_ values (μM) of munropenes A–F (**A**-**6**) from *M. pinnata* in human tumor cell lines.

Compounds	HCT116	A549	HepG2	MCF7	MDAMB
**1**	19.13	>160	>160	>160	>160
**2**	40.9	>160	>160	>160	>160
**3**	>160	>160	>160	>160	>160
**4**	17.66	>160	>160	>160	>160
**5**	57.9	>160	>160	>160	>160
**6**	32.62	>160	>160	>160	>160

## Data Availability

The data that support the findings of this study are available from the corresponding author upon reasonable request.

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
