# Peer review of "The Triterpenoids from Munronia pinnata and Their Anti-Proliferative Effects"

_molecules, 2023, doi:10.3390/molecules28196839_

Round 1

Reviewer 1 Report

Article paper entitled "The Triterpenoids and Their Anti-Proliferative Effects from Munronia pinnata " is very interesting. However before publication some issues should be clarified.

·        Results

The anti-proliferative effects of Munronia pinnata triterpenoids are not proven by the results.

·        Discussion

Discussion needs more details and improvements.

·        Conclusion

The conclusion needs to be added in the manuscript and needs to be more specific.

Minor editing of English language required

Author Response

We appreciate your kind help and meaningful comments for our manuscript. We have revised our manuscript according to your comments. The paper has been carefully revised by a professional language editing service to improve the grammar and readability. Revised portion are marked in highlighted in the paper. The main corrections in the paper are as flowing:

Comment 1: The anti-proliferative effects of Munronia pinnata triterpenoids are not proven by the results.

Response: The anti-proliferative effects has provided in the Results and Discussion Results part.

Comment 2: Discussion needs more details and improvements. 

Response: The Discussion part have be rewritten.

Comment 3: The conclusion needs to be added in the manuscript and needs to be more specific.

Response: The conclusion has added in the manuscript.

Reviewer 2 Report

A. The manuscript is incomplete. 

B. The results for the cytotoxicity assay are missing. 

C. The discussion section has the following paragraph: I do not understand this.

Authors should discuss the results and how they can be interpreted from the perspective of previous studies and of the working hypotheses. The findings and their implications should be discussed in the broadest context possible. Future research directions may also be highlighted.

Author Response

We appreciate your kind help and meaningful comments for our manuscript. We have revised our manuscript according to your comments. The paper has been carefully revised by a professional language editing service to improve the grammar and readability. Revised portion are marked in highlighted in the paper. The main corrections in the paper are as flowing:

Comment 1: The manuscript is incomplete.

Response: The manuscript has improved.

Comment 2: The results for the cytotoxicity assay are missing.   

Response: The anti-proliferative effects has provided in the Results and Discussion Results part.

Comment 3: The discussion section has the following paragraph: I do not understand this.

Response: The Discussion part have be rewritten.

Reviewer 3 Report

In the following article entitled “The Triterpenoids and Their Anti-Proliferative Effects from

Munronia pinnata” the authors Yang et. al extracted six new tirucallane-type triterpenoids, called munropenes A–F, from Munronia pinnata plants using water extraction. The compounds' structures were identified through spectroscopic methods, and their configurations were determined using various techniques. In their study, Munropenes A, B, D, and F showed anti-proliferative effects on HCT116 cell line, inhibiting its growth with IC50 values between 17.66 and 40.90 μM. Overall this manuscript is quite interesting and the data are presented well. Some minor modifications can be made prior to publication. Please see my comments below.

  1. In figure 1, in the picture section compounds are named as 1-6 whereas in the description it is written as A-F. It should be synchronized in both places.

  2. On page 9, the Discussion section is missing. Please check and modify.

  3. In figure 7, why calculated ECDs of compound 5 (which are represented as solid and dashed black lines) are so different keeping the exact same conformation of all the carbon atoms. Please check and give an explanation on this.

  4. In line 81 it is written as Phase-Sensitive HMBC (PS-COSY), please check it for a typo or explain. Line 57 can be modified as it is not clear to understand. 

  5. In the Extraction and isolation section the compounds’ characteristics IR peaks are presented. I think here the corresponding functional groups responsible can be added. Also for UV data the authors added the λmax values and possibly the corresponding molar extinction coefficient value as © which is not quite obvious for a broad spectrum of readers, so it should be mentioned. Thanks.

English language is okay, minor editing may be done.

Author Response

We appreciate your kind help and meaningful comments for our manuscript. We have revised our manuscript according to your comments. The paper has been carefully revised by a professional language editing service to improve the grammar and readability. Revised portion are marked in highlighted in the paper. The main corrections in the paper are as flowing:

Comment 1: In figure 1, in the picture section compounds are named as 1-6 whereas in the description it is written as A-F. It should be synchronized in both places.

Response: The manuscript has improved.

Comment 2: On page 9, the Discussion section is missing. Please check and modify.  

Response: The Discussion part have be rewritten.

Comment 3: In figure 7, why calculated ECDs of compound 5 (which are represented as solid and dashed black lines) are so different keeping the exact same conformation of all the carbon atoms. Please check and give an explanation on this.

Response: We are sorry for the figure 7 was wrong, this figure has been revised in the revised version.

Comment 4: In line 81 it is written as Phase-Sensitive HMBC (PS-COSY), please check it for a typo or explain. Line 57 can be modified as it is not clear to understand.

Response: We are sorry for the mistake, we has modified that mistake in in the revised version.

Comment 5: In the Extraction and isolation section the compounds’ characteristics IR peaks are presented. I think here the corresponding functional groups responsible can be added. Also for UV data the authors added the λmax values and possibly the corresponding molar extinction coefficient value as © which is not quite obvious for a broad spectrum of readers, so it should be mentioned.

Response: The Extraction and isolation section have improved according to your suggestions and comments.

Round 2

Reviewer 1 Report

Article paper entitled "The Triterpenoids and Their Anti-Proliferative Effects from Munronia pinnata ".

Based on the corrections made by the authors, this article can be accepted in its current form.

Author Response

Thank you very much for your recognition of this article.

Reviewer 2 Report

The authors have revised the manuscript and added new data for cell cytotoxicity assay. 

However, cell cytotoxicity results are not convincing. The compounds do not show cytotoxicity (four other cell lines) except in one cell line. Do others have any explanation for this?

I would suggest performing a cytotoxicity assay with a different method (Cell titer-Glo or FACS) to support and confirm the results of the CCK-8 method.

Also, add the dose-response curves for IC50 data in the manuscript.  

Author Response

We appreciate your kind help and meaningful comments for our manuscript. The dose-response curves for IC50 data have added in the Supplementary Materials according to your comments. Revised portion are marked in highlighted in the paper. However, due to time constraints, we were unable to incorporate all of your suggestions into our manuscript. Herein, we would like to give some explanations to address your inquiries. 

Comment 1: The compounds do not show cytotoxicity (four other cell lines) except in one cell line.  

Response: This is common that some compounds do not show cytotoxicity except in one cell line. In this research, compounds 1, 2, 4, and 6 exhibited moderate activities against HCT116, but do not show cytotoxicity against A549, HepG2, MCF7, and MDAMB possibly due to their specific activity profiles. However, compounds 3 and 5 also did not display any cytotoxicity (IC50>50 μM) against HCT116 cell lines, which indicated the cell cytotoxicity results could be convinced in this paper.

Comment 2: Suggesting performing a cytotoxicity assay with a different method (Cell titer-Glo or FACS) to support and confirm the results of the CCK-8 method.

Response: we are sorry for we could not performing more cytotoxicity assay due to the time constraints. The cytotoxicity assay with a different method would be carry out in further experiment for active mechanism of those compounds.

Comment 3: Add the dose-response curves for IC50 data in the manuscript.  

Response: The dose-response curves for IC50 data have added in the Supplementary Materials.